# Pitfalls in Ultrasound Diagnosis of Vascular Malformations: A Retrospective Review of 14 Nonvascular Tumors Treated as Vascular Malformations

**DOI:** 10.3390/diagnostics15040506

**Published:** 2025-02-19

**Authors:** Shintaro Mitamura, Kosuke Ishikawa, Yuki Sasaki, Naoki Murao, Satoru Sasaki

**Affiliations:** 1Department of Plastic and Reconstructive Surgery, Faculty of Medicine and Graduate School of Medicine, Hokkaido University, Sapporo 060-8638, Japan; shintaro.mitamura@huhp.hokudai.ac.jp (S.M.); sasakiyuki1120@huhp.hokudai.ac.jp (Y.S.); 2Center for Vascular Anomalies, Department of Plastic and Reconstructive Surgery, Tonan Hospital, Sapporo 060-0004, Japan

**Keywords:** angiomyoma, arteriovenous malformation, epidermal cyst, lymphatic malformation, magnetic resonance, schwannoma, sclerotherapy, vascular malformation, venous malformation, ultrasound

## Abstract

**Background/Objectives:** Vascular malformations form masses in subcutaneous and muscular tissues throughout the body and are occasionally misdiagnosed as subcutaneous nonvascular tumors. Understanding and differentiating their clinical and imaging features are crucial due to their different treatments and prognoses. This study aimed to report cases of nonvascular tumors that were initially misdiagnosed and treated as vascular malformations. **Methods:** In this retrospective observational study, we enrolled 14 (1.8%) patients with pathologically diagnosed nonvascular tumors from among 536 patients with 759 lesions of clinically diagnosed vascular malformations. **Results:** The average age at the initial visit was 41.9 years, with a male-to-female ratio of 3:11. Tumor locations included the lower limb in seven patients, the upper limb in five patients, and the trunk and head in one patient each. Ultrasound evaluation revealed 12 lesions of low-flow vascular malformations and two lesions of high-flow vascular malformations. These findings led to an initial diagnosis of venous or lymphatic malformations in 12 patients and arteriovenous malformations in two patients. Based on the clinical diagnosis, treatments administered before tumor resection included sclerotherapy in four patients and transcatheter arterial embolization in one patient. All patients underwent tumor resection. The final histopathological diagnoses included schwannoma in six patients, epidermal cyst and angiomyoma in two patients each, and other types of tumors in four patients. The average time from initial diagnosis to final histopathological diagnosis was 370 days. **Conclusions:** Multimodal diagnostic strategies, especially the use of ultrasound, enhance the differentiation between vascular malformations and nonvascular tumors.

## 1. Introduction

### 1.1. Classification of Vascular Malformations

Vascular malformations, which arise from congenital differentiation and dysplasia of arteriovenous and lymphatic vessels, form masses within subcutaneous and muscular tissues throughout the body [1]. According to the International Society for the Study of Vascular Anomalies classification, vascular malformations include a range of vascular anomalies, including both vascular tumors and malformations. These are classified into venous malformations (VMs), lymphatic malformations (LMs), arteriovenous malformations (AVMs), capillary malformations, and mixed vascular malformations [2].

#### 1.1.1. VMs

VMs are the most common vascular malformations, accounting for 44–64% of cases [3]. They are hemodynamically inactive, low-flow malformations occurring on the aggregate side of the vascular network. VMs are characterized by enlarged and distorted venous tracts with walls that lack smooth muscle cells and are irregularly lined with a flat, continuous layer of endothelial cells. These malformations usually present at birth as small bluish spots or plaques or as a network of dilated veins visible in the skin, and they tend to enlarge gradually over the patient’s lifetime. When the lesion is deep, the overlying skin may appear normal. VMs are generally elastic soft subcutaneous masses but may involve the skin, mucous membranes, and soft tissues, especially muscles, joints, bones, and even internal organs [1,3].

#### 1.1.2. LMs

LMs are malformations of the lymphatic system, characterized by the presence of small vesicles or large pouches filled with lymphatic fluid. The most common anatomical location is the head and neck (36.5%), followed by the extremities and axilla (31%) and the trunk (24.1%) [4].

Combined microcystic and macrocystic forms are relatively common. Microcystic LMs infiltrate soft tissues, including the skin and mucosa, leading to the emergence of clear or hemorrhagic vesicles. These malformations can also affect visceral regions in the thorax, abdomen, and even bones. Macrocystic LMs form large translucent subcutaneous lumps. Upon puncture aspiration, these macrocystic formations yield clear yellow lymphatic fluid. LMs are harder than VMs and do not compress under pressure. However, they can abruptly enlarge in response to local inflammation or bleeding [5].

#### 1.1.3. AVMs

AVMs are hemodynamically active, high-flow vascular malformations. These consist of a nidus made up of arterial feeders and enlarged draining veins, which are directly connected through micro- and macro-fistulas. AVMs can occur in both superficial and visceral locations. The head and neck are the most commonly affected areas, though AVMs can also occur in any location. The age of onset varies: 34% of AVMs are present at birth, 21% manifest during childhood, and 8.5% emerge during adolescence [6]. According to the Schobinger classification, the first stage of AVM development is characterized by faint redness on the superficial skin. The second stage involves symptoms such as a burning sensation and a palpable thrill. In the third stage, rest pain, skin ulcers, and bleeding occur, and in the fourth stage, high-output heart failure is observed [7].

### 1.2. Diagnostic Tools to Differentiate Vascular Malformations from Nonvascular Tumors

Vascular malformations are typically present at birth, but they may enlarge gradually and are often identified only after bleeding or other clinical events occur. Thus, they are sometimes misdiagnosed as nonvascular tumors, especially when they present as normal-colored subcutaneous tumors.

Ultrasound (US) is the most important diagnostic modality and should be performed in all patients in routine practice. Several imaging combinations can simplify the diagnosis of subcutaneous tumors. High-frequency linear array probes are the best tool for evaluating superficial lesions. For optimal visualization of small vessels and the detection of low-velocity arterial and venous flows, low-flow color Doppler settings are recommended. Color Doppler imaging also facilitates the placement of the Doppler gate for spectral waveform analysis. In Doppler studies, a low-resistance waveform is characteristic of inflammatory and neoplastic processes and is indicative of high-flow states or arteriovenous shunting [8]. A flat low-velocity waveform that increases with flow augmentation techniques helps diagnose VM. A water bath or standoff technique is helpful for visualizing superficial lesions [9]. The advantages of US in diagnosing vascular malformations include its ability to quickly and noninvasively identify the speed, direction, and type of flow. However, its disadvantages include a limited field of view and penetration depth and the difficulty in depicting flat, superficial lesions with low-flow vessels. For instance, low-flow vascular malformations such as VM and LM can produce weak blood flow signals, leading to potential misdiagnosis as schwannomas [5,10]. Similarly, AVM can easily be mistaken for hypervascular tumors, including malignancies [11], because of their abundant and rapid blood flow signals. Magnetic resonance (MR) is also an important diagnostic modality. Vascular malformations are generally characterized by low signal intensity on T1-weighted imaging (T1WI) and high signal intensity on T2-weighted imaging (T2WI). However, MR has a significant limitation: distinguishing vascular malformations from nonvascular tumors with similar signal characteristics can be challenging. Computed tomography (CT) is useful in depicting the delineation of the lesion and its relationship to the surrounding tissue and bone.

Vascular malformations and other nonvascular tumors are occasionally misdiagnosed [5,10,11,12,13,14,15,16] despite detailed patient histories, thorough physical examinations, and advanced imaging techniques because of this clinical and imaging mimicry. Biopsy is not a commonly used diagnostic method [1]. The main treatment for vascular malformations includes sclerotherapy or transcatheter arterial embolization (TAE). Sclerotherapy is particularly effective for managing unresectable vascular malformations, as it helps reduce massive bleeding, lymphatic leakage, and post-treatment re-expansion. However, inappropriate application of sclerotherapy to nonvascular tumors can delay accurate treatment and, in cases of malignancy, result in tumor cell seeding. Differentiating between vascular malformations and nonvascular tumors is therefore critical, as their treatment approaches and prognoses differ significantly. Despite the importance of accurate differentiation, reports addressing this issue remain scarce. Therefore, in this study, we aimed to report a series of 14 patients who were clinically diagnosed with and treated for vascular malformations but were pathologically diagnosed with nonvascular tumors and to review the key diagnostic features and imaging findings of vascular malformations and nonvascular tumors.

## 2. Materials and Methods

A total of 536 patients with 759 lesions of vascular malformations that were clinically diagnosed in the Department of Plastic and Reconstructive Surgery, Tonan Hospital, between July 2008 and September 2023, were identified from the files of the Department of Diagnostic Pathology. All patients underwent tumor resection and histopathological examination. Of these lesions, 745 lesions of vascular malformations with an initial clinical diagnosis consistent with the final histopathological diagnosis were excluded. Patients who were clinically diagnosed with vascular malformations but were pathologically diagnosed with nonvascular tumors were included in this study. Clinical data, including the initial diagnosis by previous doctors and/or those in our hospital, treatments before tumor resection, and medical history, were extracted from medical records. Cases 10 and 14 have been previously reported by our group [17].

## 3. Results

### 3.1. Clinical Features

Fourteen consecutive patients treated for vascular malformations were enrolled in this study (Table 1). The cohort comprised 3 men and 11 women, with an average age of 41.9 ± 24.2 (standard deviation) years (range, 2–82). The initial symptoms were lumps in all patients and pain in 11. The tumors were located in the lower limbs in seven patients, the upper limbs in five patients, and the trunk and head in one patient each. Comprehensive examinations, including US and MR, were conducted for all patients. CT and CT angiography were performed in seven patients. Digital subtraction angiography (DSA) during TAE was conducted in one patient. Two patients underwent biopsies performed previously by physicians.

US revealed 12 lesions of low-flow vascular malformations and two lesions of high-flow vascular malformations. Plain MR was performed in two patients, and contrast-enhanced MR in 12 patients. Among these, two lesions showed no contrast enhancement, six lesions showed partial enhancement, and four lesions showed marked enhancement. The initial clinical diagnoses were made at our hospital based on the findings from these imaging modalities. Diagnoses included VM in 11 patients, AVM in 2 patients, and LM in 1 patient. Based on these clinical evaluations, the following treatments were performed in our hospital: incisional biopsy in 2 patients, sclerotherapy in 4 patients, and TAE in 1 patient. Due to the lack of lesion reduction following these interventions, total tumor resection was performed based on the clinical diagnosis of vascular malformations in all patients. The final histopathological diagnosis was schwannoma in six patients, epidermal cyst and angiomyoma in two patients each, and angiofibroma of soft tissue and traumatic neuroma in one patient each. Patients in cases 13 and 14 were diagnosed with non-benign tumors, extraskeletal myxoid chondrosarcoma (malignant), and solitary fibrosis tumor (intermediate malignant), respectively. The average time from the initial clinical diagnosis to the final histopathological diagnosis was 370 ± 539 (standard deviation) days (range, 14–1893). No recurrence was observed during the follow-up period, except for in case 14. Case 14 involved a 70-year-old man with a right thigh mass present for approximately 50 years, which had gradually increased in size and showed calcification. The patient underwent surgery for tumor reduction under a clinical diagnosis of AVM. The histopathological diagnosis was solitary fibrous tumor. Three months after surgery, the patient died of aggressive tumor re-expansion and pulmonary metastasis. The six other representative cases are described below.

### 3.2. Case 1

A 2-year-old girl presenting with swelling in her right hand, reportedly caused by her mother, was referred to our department. The patient had no relevant medical or family history. Examination revealed an elastic, soft, and non-tender mass palpable on the ulnar side, extending from the right hand to the forearm (Figure 1a). US revealed a hypoechoic area (Figure 1b) with a poor blood flow signal (Figure 1c). MR showed an isointense mass without enhancement on gadolinium (Gd)-enhanced T1WI (Figure 1d) and a hyperintense mass extending from the right hand to the forearm on short τ inversion recovery (STIR) (Figure 1e,f). She was diagnosed with a VM, and sclerotherapy with 3% polidocanol foam was performed four times (Figure 1g,h, Appendix A). However, the mass did not shrink. Five years after the initial diagnosis, tumor resection was undertaken (Figure 1i). The histopathological diagnosis was schwannoma.

### 3.3. Case 2

A 15-year-old girl presented with a mass in the right buttock that had been present for 5 years. Her medical history included a right buttock contusion sustained 5 years before presentation. US revealed a subcutaneous, homogeneous, hypoechoic cyst that did not deform under probe compression (Figure 2a). MR showed a 5 cm irregularly shaped, homogeneous mass on STIR (Figure 2b), with Gd-enhancement (Figure 2c). A palpable, elastic, firm and mildly tender mass was present in the right buttock (Figure 2d). She was diagnosed with a VM extending into the gluteus maximus, accompanied by a thrombus likely caused by trauma. Treatment consisted of three courses of sclerotherapy (Appendix A). After sclerotherapy, US revealed an obscured cyst with no blood flow signal (Figure 2e), and MR showed a hypointense mass on STIR (Figure 2f,g). However, owing to residual lesions and persistent tenderness, the lesion was excised three years after the initial clinical diagnosis (Figure 2h). The histopathological diagnosis was schwannoma.

### 3.4. Case 3

A 25-year-old woman presented with a progressive, painless subcutaneous mass in the occipital region that had been present for 5 years. A soft, elevated mass measuring 30 mm was palpable in the right occipital region. MR showed a well-defined hypo- to isointense mass with a fluid–fluid level on T2WI (Figure 3a) and heterogeneous enhancement on Gd-enhanced T1WI (Figure 3b,c). An incisional biopsy performed by a previous physician resulted in intraoperative leakage of old blood components from the mass, leading to an initial diagnosis of VM. The patient was referred to our department for further treatment. At her initial visit, US showed a hypoechoic mass with poor blood flow and an enhanced posterior echo (Figure 3d). Total tumor resection was selected for cosmetic reasons. The histopathological diagnosis was schwannoma. The excised specimen showed bleeding on both the superficial and deep sides of the tumor (Figure 3e). Sampling of only superficial tissues in the initial biopsy focused on the hemorrhage and VM-like structures, which may have led to an incorrect initial diagnosis of VM.

### 3.5. Case 4

A 39-year-old woman who had been aware of a progressive and painful mass on her right upper arm for the past 9 years was suspected of having an intramuscular VM based on a needle biopsy performed by her previous physician (Figure 4a). The patient was referred to our department for sclerotherapy or excisional treatment. A soft, elevated, and tender mass, which was 6 × 5 cm in size, was palpable in the right upper arm. US revealed a heterogeneous mass with partial high-flow signals and enhanced posterior echoes (Figure 4b–d). MR showed a well-defined mass with an isointense-to-hyperintense signal on STIR (Figure 4e) and heterogeneous enhancement of the lesion on Gd-enhanced T1WI, which appeared to be a drainage vein (Figure 4f). Based on these findings, a diagnosis of intramuscular VM was made, and sclerotherapy with 3% polidocanol foam was performed. However, three-dimensional CT angiography as an additional examination, failed to identify any inflow or outflow vessels (Figure 4g,h), and a re-biopsy was performed due to mass growth, resulting in a hematoma. One year after the initial clinical diagnosis, a total tumor resection was performed. The tumor was a degenerated cyst, 4 × 3 cm in size, covered with multiple capsules, and adhered to the surrounding tissue, including the periosteum (Figure 4i). The histopathological diagnosis was degenerated schwannoma.

### 3.6. Case 7

A 35-year-old woman presented with a progressive and painful subcutaneous mass in the right thigh that had persisted for 5 years (Figure 5a). US revealed a heterogeneous hypoechoic cyst without a blood flow signal or deformation upon probe compression (Figure 5b). MR showed a well-defined dilated luminal structure, 5.6 × 1.4 cm in size, a hyperintense signal on STIR, and no contrast effect on Gd-enhanced T1WI (Figure 5c). She was diagnosed with LM with hemorrhage and treated with two courses of sclerotherapy. However, owing to residual lesions (Figure 5d), tumor resection was performed. The excised tumor showed adhesions to the surrounding tissues, including the deep fascia, suggesting post-inflammatory changes due to sclerotherapy. The histopathological diagnosis was an epidermal cyst.

### 3.7. Case 9

A 66-year-old man was referred to our department with complaints of swelling and enlargement of the base of the right ring finger and loss of sensation in the finger apex. An elastic, soft, painless mass was found in the fourth metacarpal of the palm (Figure 6a). US revealed a heterogeneous hypoechoic area with a relatively high-flow blood signal (Figure 6b). MR showed a hypo-to-iso-intense signal on T1WI, hyperintense signal on STIR, and heterogeneous contrast effect on Gd-enhanced T1WI (Figure 6c), while three-dimensional CT angiography showed a well-defined homogenous hypointense lesion with enhancement from the center of the tumor in the early arterial phase (Figure 6d), which failed to identify any inflow or outflow vessels. The tumor was diagnosed as an AVM, and tumor resection was selected because of the localized lesion (Figure 6e). The histopathological diagnosis was angiomyoma.

## 4. Discussion

We examined 14 patients who were clinically diagnosed with and treated for vascular malformations but were pathologically diagnosed with nonvascular tumors. This study is the largest case series reporting mimickers of vascular malformations other than two reports of 11 cases each [13,18]. In the former report, 11 patients who were initially treated for VM were pathologically diagnosed with spindle cell hemangiomatosis, a vascular tumor that is easily confused with vascular malformations [13]. In the latter report, three patients who were initially misdiagnosed with VMs received a final diagnosis of sarcoma [18]. In the present study, the final histopathological diagnosis was schwannoma, which was the most common, occurring in six cases, with sarcoma in one case. US and MR findings used to differentiate vascular malformations and nonvascular tumors are described below.

### 4.1. US, MR, and CT Findings of Vascular Malformations

#### 4.1.1. VMs

On US, VMs exhibit a variety of shapes, which can be categorized into three main venogram patterns: cavernous, spongiform, and dysmorphic [19]. On grayscale US, superficial VMs are compressible and display a heterogeneous echotexture (98%). They may appear hypoechoic (82%), hyperechoic (10%), or isoechoic (8%) compared with surrounding structures [20]. Based on venous drainage patterns, four types of VMs have been proposed: type I, isolated malformations without venous drainage; type II, malformations draining into normal veins; type III, malformations draining into dysplastic veins; and type IV, malformations primarily consisting of venous ectasia [21,22]. On color Doppler US, 78% of VMs demonstrate monophasic flow waveforms, 16% exhibit minimal or no flow waveforms, and 6% present with biphasic flow waveforms. Minimal or absent flow waveforms may indicate extremely low flow rates below the detectable limit or thrombosis, potentially leading to diagnostic challenges [20]. Biphasic flow waveforms might reflect mixed capillary components. The most important feature of US in VMs is the lesion’s response to compression. When compressed with a probe, the cavity rapidly empties and refills once the pressure is released. Additionally, phleboliths, present in 16% of VMs, are strong indicators. These appear as concentric calcifications on imaging and produce acoustic shadows [20].

MR findings include a focal or diffuse hyperintense signal on T2WI, often comprising multiple spaces separated by the septum. In intramuscular VMs, the spaces can be seen as channels that are parallel to the intervening muscle fibers. Small, subtle fluid-fluid levels may be observed [23]. Phleboliths are the most diagnostically valuable finding and are seen as round no-signal areas called “dot signs” [24].

On plain CT, soft tissue density lesions can be visualized, with phleboliths appearing as rounded high densities. Depending on the degree of blood congestion within VMs, VMs are variously enhanced by contrast administration [19,25].

#### 4.1.2. LMs

On US, the classic appearance of an LM is that of multiloculated cystic masses with imperceptible walls that are anechoic in their natural state and offer posterior acoustic enhancement. The appearance of a lesion depends on the size of the cystic space. Macrocystic LM shows a unilocular or multilocular cystic mass that sometimes contains echogenic debris with an intervening stroma, and the entire area is hyperechoic with internal hemorrhage. LMs are rarely unilocular; in fact, a unilocular cystic mass is unlikely to be an LM. On color Doppler US, intralesional retracted clots appear as solid areas of varying echogenicity without internal vascularity [22,26,27]. Microcystic LMs may be hyperechoic because of the number of interfaces. Color Doppler US indicates arterial and venous flows within the septum and stroma, which correspond to the histological presence of arteries and veins within the cyst walls and stroma. Particularly in the microcystic type, the lesion does not regress under pressure from the probe due to its elasticity [28,29]. Pure LMs lack associated soft tissue components and exhibit no internal flow. Additionally, perilesional inflammation is absent unless an active infection is present. In cases of infection, debris or echogenic fluid may be observed, accompanied by the development of hyperemia along the capsule and septum [30]. During US-guided sclerotherapy, lymphatic fluid can be obtained by puncture aspiration from macrocystic LMs but not from nonvascular tumors.

MR shows that LMs are well characterized as hyperintense on T2WI. However, if complicated by hemorrhage or infection, the signal intensity varies depending on the protein content and presence of blood products. After contrast administration, the cyst walls and septa are enhanced, whereas the central part of the cyst is not. Large and/or anomalous adjacent venous channels are often observed, and the LM is often seen communicating with adjacent veins [23,30]. The presence of fluid–fluid levels, lack of enhancement, and absence of phleboliths differentiate LMs from VMs [26]. Microcystic LMs are more difficult to differentiate from tumors because the cystic spaces may be so small that the lesions appear more diffusely enhanced [8].

On plain CT, LMs are seen as low-density lesions. On post-contrast scans, the cyst walls and septum are enhanced, but the fluid contained within the cystic space is not. CT is the best tool for demonstrating bony changes due to LMs [9].

#### 4.1.3. AVMs

Grayscale US shows multiple dilated tortuous vascular channels diffusely involving subcutaneous soft tissues. On Color Doppler US, the lesions show arterial and venous flows with high vessel density. The arterial and venous systems are in direct contact; therefore, the resistance index indicates a low resistive flow. However, hypervascularized tumors with arterial blood supply may show similar findings and should be treated carefully. Hypervascular solid tumors are tumors accompanied by numerous blood vessels and are distinct from vascular malformations, lesions in which the vascular vessels themselves are abnormally formed. Vascular malformations are divided into low-flow and high-flow types based on the velocity of flow within the lesion.

MR is useful for understanding the three-dimensional structures of the feeding artery and draining vein. The characteristically enlarged arterial components appear as flow voids on T1WI and T2WI. Gd-enhanced MR angiography shows good enhancement and the appearance of vessel clusters. Three-dimensional CT angiography is useful for understanding the relationship between feeding arteries and draining veins. DSA is used to distinguish between the two. It is a characteristic of the arterial phase to show the drainer, reflecting early venous perfusion [31].

### 4.2. Mimickers of Vascular Malformations

#### 4.2.1. Schwannoma (Neurinoma, Neurilemmoma)

Schwannomas are benign tumors derived from Schwann cells. They are usually solitary but multiple in neuromatosis. The masses touch intradermally or subcutaneously in the form of hard elastic spheres or beads. They are often tender and radiating. They rarely become malignant and are called malignant peripheral nerve sheath tumors.

Grayscale US shows a well-defined, spherical, or lobulated hypoechoic mass with heterogeneous echoes and enhanced posterior echoes. Careful observation confirms continuity with the peripheral nerves, the thin tail-like hypoechoic part from the nerve trunk, which was connected proximally or distally to one or both sides of the tumor [32]. Color Doppler US shows abundant blood flow signals within the lesion. Yang et al. reported that 49 of the 54 schwannomas were vascularized to varying degrees, and 22 were parallel and close to a vessel [32]. In a report of 49 schwannomas by Tsai et al., all of them showed hypervascular changes, and of these, 47 had peripheral and central vascularity, and 2 had peripheral vascularity [33]. Considering that nerves and vessels often run parallel to each other, especially in the extremities, a tumor that is adjacent to a vessel is more likely to be a schwannoma.

On MR, the classic appearance of a schwannoma is that of an iso-to-hypointense signal on T1WI, a hyperintense signal on T2WI, and post-contrast enhancement. Heterogeneous signal intensity and postcontrast enhancement are suggestive of internal hemorrhagic and myxoid/cystic changes [34]. The entering and exiting nerve signs describe the presence of a peripheral nerve that enters and exits a mass. This may or may not be observed on imaging, depending on mass size. The target sign consists of a centrally low/intermediate signal and a peripherally high signal on T2WI [34,35]. Therefore, Gd-enhanced MR in the early and delayed phases may be useful for differentiation from VMs.

Olivieri et al. first listed schwannomas as mimickers that can be confused with VMs [10]. Notably, they are hypervascularized tumors, and when complicated by hemorrhage, imaging findings can be confused with those of VMs. Tenderness or radiating pain may be supportive findings for schwannomas.

#### 4.2.2. Epidermal Cyst

Epidermal cysts are the most common subcutaneous tumors [36]. They adhere to the epidermis, and in typical cases, the skin has a normal color, is relatively firm, and has a central orifice. When infected, they become tender and often self-destruct and drain themselves.

On grayscale US, the mass is hypoechoic, with a well-defined border bordering the dermis and an enhanced posterior echo, and the typical findings are similar to those of a normal testis. A hypoechoic area contiguous with the epidermis may be observed; however, in many cases, no pits are present. Internal echogenicity is variable and is not helpful in diagnosis. If the cyst wall ruptures, it exhibits an irregular morphology. Color Doppler US shows no blood flow signal, whereas with infection, the blood flow signal around the cyst increases, reflecting inflammation, which is called a twinkle artifact [37].

On MR, epidermal cysts are depicted as a unifocal cyst with a thin capsule in contact with the skin, with an isointense signal on T1WI and a hyperintense signal on T2WI, and the contrast effect is enhanced only in the capsular-like structure. They are characterized by a low apparent diffusion coefficient obtained from diffusion-coordinated images. White et al. reported epidermal cysts as mimickers that could be confused with LMs [5].

#### 4.2.3. Angiomyoma (Vascular Leiomyoma, Angioleiomyoma)

Angiomyoma is a benign tumor of the musculoskeletal system derived from the vascular smooth muscle, which occurs in the subcutis or deep dermis and may contain adipose tissue. The most common sites are the lower extremities. It is characterized by tenderness and pain elicited by cold stimuli [38].

US findings show a subcutaneous solid and homogeneous tumor with smooth margins and an average size of 11 mm [39]. The internal echo is uniform, and the posterior echo is enhanced. Color Doppler US typically shows an abundant blood flow signal, but some reports indicate that it is undetectable in approximately 60% of cases [40]. According to a report by Koga et al., 3 of 14 cases involved calcified tumors, which could lead to a misdiagnosis [39].

MR depicts it as a well-defined subcutaneous mass and shows an isointense signal on T1WI and an isointense to hyperintense signal on T2WI, with strong enhancement after contrast administration [38,41]. The differentiation points between vascular malformations and mimicking nonvascular tumors are summarized in Table 2.

### 4.3. Lessons from Our Cases

Some lessons learned from the aforementioned case series are outlined. If differentiation is difficult based on imaging findings, a biopsy should be performed to make a histopathological diagnosis, and the patient should be managed with appropriate measures to control bleeding. In fact, in cases 3 and 4, sampling only the superficial layer of the tumor resulted in focusing on the hemorrhage and VM-like structures, which may have led to an incorrect initial diagnosis of VM. The results of partial biopsy should not be overstated. An incisional biopsy requires an adequate specimen and should be performed by an experienced specialist. Once treatment is started but the expected therapeutic effect is not seen, or if there are problems with the clinical course, the diagnosis should be reviewed.

If sclerotherapy is misapplied to a nonvascular tumor, the tumor may become inflamed along with the surrounding tissue and adhere to the subcutaneous tissue, muscle, and bone, making total excision more difficult. In fact, in cases 1, 2, and 7, extra tissue was removed because of the adhesion of the tumors to the surrounding tissue. In case 4, the tumor was strongly attached to the periosteum of the humerus and was difficult to remove. If interventions (including sclerotherapy and TAE) are misapplied to malignant tumors, there is a risk of disease activation, which promotes invasion and metastasis. In case 14, the final histopathological diagnosis was solitary fibrosis tumors; however, it could be confused with AVMs and should be treated with caution. Hypervascular malignant tumors and AVMs should be differentiated with caution. Scorletti et al. reported that over a 12-year period, 11 patients were initially treated for vascular malformations that were later found to be malignant soft tissue tumors, two of whom died [18]. Dermatofibrosarcoma protuberans, classified as an intermediate malignancy, should also be considered. Torresetti et al. reported a case in which a patient was misdiagnosed as having a “hemangioma” and treated accordingly, only to be diagnosed with dermatofibrosarcoma protuberans some 30 years later [14].

The role of color Doppler US in differentiating benign lesions from malignant tumors has been evaluated with limited success. Several studies have shown that an organized vascular pattern is more common in benign soft tissue masses, whereas a chaotic vascular pattern with randomly distributed tortuous vessels showing abrupt changes in caliber is more common in malignancy [30]. LMs are multisegmented masses that are expected to have minimal or no peripheral and septal vascularity, which may be accentuated during periods of hemorrhage or infection. Considerable overlap between benign and malignant lesions was found when examining the mean end-diastolic volume and mean resistance index [42]. Therefore, the clinical value of Doppler US in ruling out malignancies is limited.

This study has some limitations. First, some cases may have been omitted due to manual case searching. Second, this study was conducted at a single institution with a relatively small number of cases; therefore, a multi-institutional study is warranted in the future.

## 5. Conclusions

The diagnosis of vascular malformations mistaken for nonvascular tumors can lead to harmful treatment, and a better understanding of this condition is essential. Multimodal strategies with imaging modalities, including US and MR, in addition to clinical information, help differentiate vascular malformations from nonvascular tumors. If treatment is started without a histopathological diagnosis but the expected therapeutic effect is not seen, the histopathological diagnosis should be confirmed.

## Figures and Tables

**Figure 1 diagnostics-15-00506-f001:**
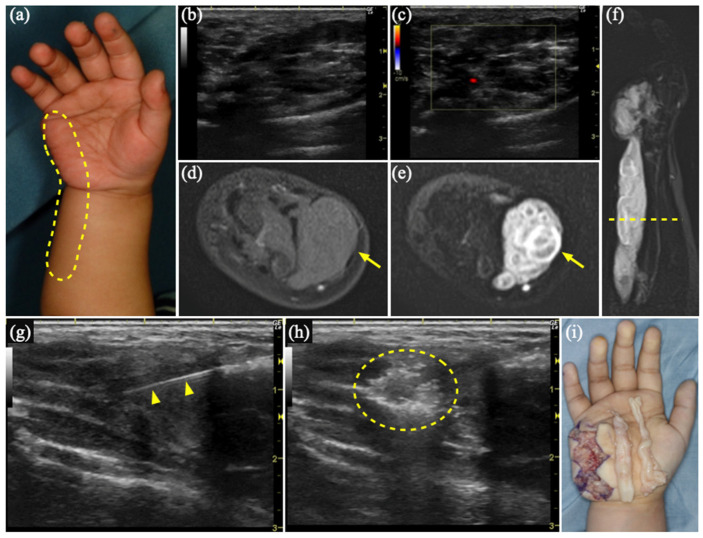
Case 1 of a schwannoma mimicking a venous malformation. (**a**) An elastic, soft, and non-tender mass on the ulnar side from her right hand to her forearm (dashed circle). (**b**) Grey-scale ultrasound (US) shows a hypoechoic area. (**c**) Color Doppler US shows poor blood flow signal within the lesion. (**d**) Magnetic resonance (MR) reveals an isointense mass without enhancement on gadolinium (Gd)-enhanced T1-weighted imaging (arrow). (**e**) MR shows a hyperintense mass on short τ inversion recovery (arrow), (**f**) which extends from her right hand to her forearm (the dotted line indicates the cross-section of (**d**,**e**). (**g**) US-guided sclerotherapy using foamed 3% polidocanol: inserting a needle (arrowheads) into the hypoechoic cyst; and (**h**) once the injection was administered, a hyperechoic area appears (dashed circle). (**i**) Intraoperative photograph shows a white tone, cord-like tumor on the ulnar side of her right hand.

**Figure 2 diagnostics-15-00506-f002:**
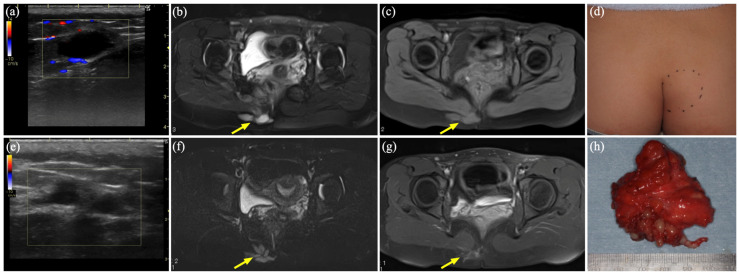
Case 2 of a schwannoma mimicking a venous malformation. (**a**) Color Doppler ultrasound (US) shows a hypoechoic area with numerous fine blood flow signals in the surrounding tissue. (**b**) Magnetic resonance (MR) shows a hyperintense mass on short τ inversion recovery (STIR) (arrow). (**c**) MR shows a hypointense mass without enhancement on gadolinium (Gd)-enhanced T1-weighted imaging (T1WI) (arrow). (**d**) An elastic, hard, and tender mass in the right buttock (dashed circle). (**e**) Color Doppler US shows cysts with reduced and obscured borders, as well as reduced blood flow signals in the surrounding tissue. (**f**) MR shows the shrunken mass on STIR (arrow). (**g**) MR shows the shrunken mass without enhancement on Gd-enhanced T1WI (arrow). (**h**) An intraoperative photograph shows a resected specimen measuring 5 × 3.5 cm.

**Figure 3 diagnostics-15-00506-f003:**
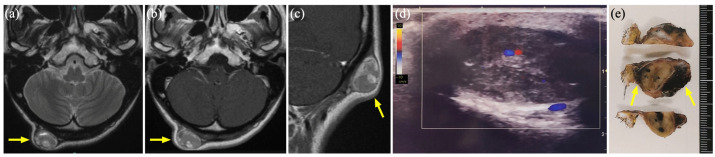
Case 3 of a schwannoma mimicking a venous malformation. (**a**) Magnetic resonance (MR) shows well-delineated masses with fluid–fluid levels on T2-weighted imaging (arrow). (**b**,**c**) MR reveals heterogeneous enhancement on gadolinium-enhanced T1-weighted imaging (arrows). (**d**) Color Doppler ultrasound shows a hypoechoic cyst with poor blood flow signals and enhanced posterior echo. (**e**) Formalin-fixed specimens have hematomas on both the superficial and deep sides of the body (arrows).

**Figure 4 diagnostics-15-00506-f004:**
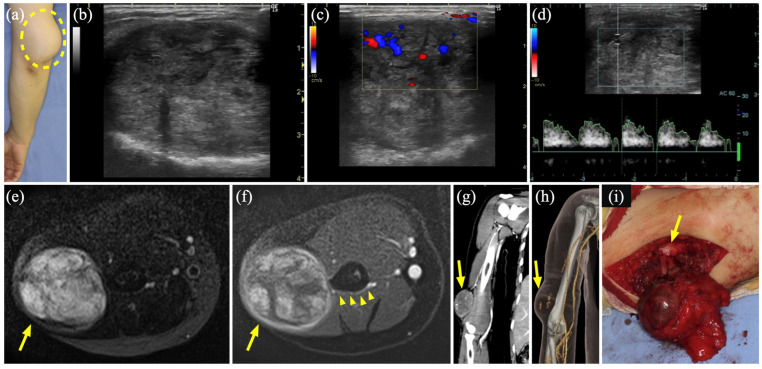
Case 4 of a degenerated schwannoma mimicking a venous malformation. (**a**) An elastic, soft, and tender mass on the right upper arm (dashed circle) (**b**) Greyscale ultrasound (US) shows a heterogeneous mass. (**c**) Color Doppler US reveals partial high-flow signals. (**d**) Doppler US shows that the waveform peaked at 18 cm/s. (**e**) Magnetic resonance (MR) shows the iso-to-hyperintense mass on short τ inversion recovery (arrow). (**f**) MR shows a heterogeneous mass (arrow) with a drainage vein (arrowheads) on gadolinium-enhanced T1-weighted imaging. (**g**) Enhanced computed tomography (CT) shows the heterogeneous mass (arrow). (**h**) Three-dimensional CT angiography shows no obvious inflow or outflow vessels (arrow). (**i**) Intraoperative photograph shows a 6.5 × 5.0 × 4.0 cm specimen, which adheres to the periosteum of the upper humerus (arrow).

**Figure 5 diagnostics-15-00506-f005:**
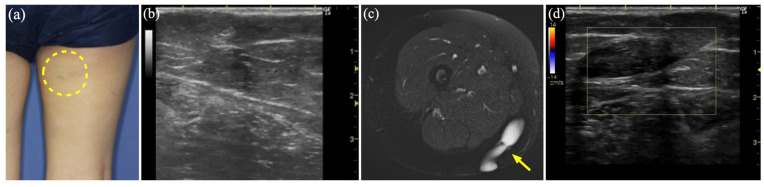
Case 7 of an epidermal cyst mimicking a lymphatic malformation. (**a**) A 4 × 2 cm large tender mass on the right thigh (dashed circle). (**b**) Greyscale ultrasound (US) shows a hypoechoic area with slightly enhanced posterior echo. (**c**) Magnetic resonance imaging shows 5.6 × 1.4 cm hyperintense large dilated luminal structures with well-defined boundaries on short τ inversion recovery (arrow). (**d**) Color Doppler US shows a residual cyst with no blood flow.

**Figure 6 diagnostics-15-00506-f006:**
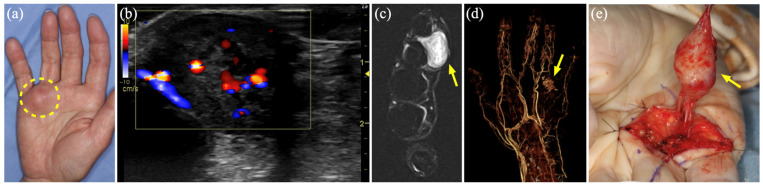
Case 9 of an angiomyoma mimicking an arteriovenous malformation. (**a**) A 3 × 2.5 cm non-tender mass at the base of the ring finger of the right hand (dashed circle). (**b**) Color Doppler ultrasound shows a hypoechoic area with a relatively high-flow blood signal. (**c**) Magnetic resonance imaging shows a hyperintense mass on short τ inversion recovery (arrow). (**d**) Three-dimensional computed tomography angiography shows a hypervascular mass (arrow). (**e**) Intraoperative photograph shows a specimen about to be resected (arrow).

**Table 1 diagnostics-15-00506-t001:** Clinical features of the 14 nonvascular tumors mimicking vascular malformations.

Case No.	Age, Years *	Sex	Tumor Location	Initial Clinical Diagnosis	Imaging Modalities	Treatment Before Tumor Resection	Final Histopathological Diagnosis	Days to Final Diagnosis
1	2	F	Rt hand, forearm	VM	US, MR	Sc, 4 times	Schwannoma	1893
2	15	F	Rt buttock	VM	US, MR, CT	Sc, 2 times	Schwannoma	1186
3	25	F	Lt head	VM	US, MR	Biopsy	Schwannoma	37
4	39	F	Rt upper arm	VM	US, MR, 3D-CTA	Sc, 1 time 2 biopsies	Schwannoma	788
5	52	F	Rt forearm	VM	US, MR	Biopsy	Schwannoma	55
6	31	F	Rt lower leg	VM	US, MR	None	Schwannoma	343
7	35	F	Rt thigh	LM	US, MR, CT	Sc, 2 times	Epidermal cyst	440
8	29	F	Lt lower leg	VM	US, MR	None	Epidermal cyst	74
9	66	M	Rt hand	AVM	US, MR, 3D-CTA	None	Angiomyoma	19
10	82	M	Lt lower leg	VM	US, MR, 3D-CTA	None	Angiomyoma	75
11	50	F	Rt foot	VM	US, MR, CT	None	Angiofibroma of soft tissue	14
12	13	F	Rt thumb	VM	US, MR	None	Traumatic neuroma	33
13	78	F	Rt lower leg	VM	US, MR	None	Extraskeletal myxoid chondrosarcoma	26
14	70	M	Rt thigh	AVM	US, MR, 3D-CTA, DSA	TAE, 2 times	Solitary fibrous tumor	199

* Age (years) at initial visit; AVM, arteriovenous malformation; CT, computed tomography; DSA, digital subtraction angiography; F, female; Lt, left; LM, lymphatic malformation; M, male; MR, magnetic resonance; Rt, right; Sc, sclerotherapy; 3D-CTA, 3D-CT angiography; US, ultrasound; TAE, transcatheter arterial embolization; VM, venous malformation.

**Table 2 diagnostics-15-00506-t002:** Differentiation points between vascular malformations and mimicking nonvascular tumors.

		Vascular Malformations				Mimicking Nonvascular Tumors		
		VM	LM		AVM	Schwannoma	Epidermal Cyst	Angiomyoma
			Macrocystic	Microcystic				
Clinical findings	Skin color	Blue ~ purple *	Normal	Yellow ~ red vesicles *	Red ~ purple *	Normal	Normal	Normal ~ brown
	Tenderness	− ~ +	− ~ +		− ~ +	+	−	+
	Boundary	Clear ~ unclear	Clear	Unclear	Clear ~ unclear	Clear	Clear	Clear
	Pulsation	−	−		+	−	−	+
Needle aspiration		±	+	−	±	−	−	−
US	Form	Polycystic	Cystic	Polycystic	Dilated and tortuous blood vessels	Solid	Cystic	Solid
	Flow	Low-flow	No-flow		High-flow	Hypervascular	No-flow	Hypervascular
	Compression deformation	+	−		−	−	−	−
MR	T1WI	Low ~ iso	Low		Flow void	Low	Iso	Iso
	T2WI	High	High		Flow void	Iso at center, high at margin	High	High
	Contrast effect	+	−		++	++ at center, + at margin	+ at margin	++
Other features		Phleboliths	Fluid–fluid level due to hemorrhage or infection				Low apparent diffusion coefficient on DWI	

* Normal in case of deep lesions; AVM, arteriovenous malformation; DWI, diffusion weighted imaging; LM, lymphatic malformation; MR, magnetic resonance; US, ultrasound; VM, venous malformation; WI, weighted imaging.

## Data Availability

The data supporting the findings of this study are available from the corresponding author upon reasonable request.

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
