# Peer review of "Pitfalls in Ultrasound Diagnosis of Vascular Malformations: A Retrospective Review of 14 Nonvascular Tumors Treated as Vascular Malformations"

_diagnostics, 2025, doi:10.3390/diagnostics15040506_

Round 1
Reviewer 1 Report
Comments and Suggestions for Authors
General comment
This study provides a comprehensive retrospective analysis of cases where nonvascular tumors were initially misdiagnosed as vascular malformations. The findings are clinically significant and contribute to improving the accuracy of ultrasound diagnosis. The manuscript is well-structured, and the imaging descriptions are informative.
Specific comment
In Table 2, please discuss differences between "high-flow" in arteriovenous malformation and "hypervascular" in angiomyoma.
Do you have specific differential points or parameters during Doppler ultrasound?
This differential diagnosis will be the key point of this manuscript. Please discuss this point more in detail.
Comments on the Quality of English LanguageGrammar and Style: Some sentences are lengthy and could be simplified for better readability.
Reviewer 2 Report
Comments and Suggestions for Authors
Clinical presentation of disease in Introduction should be included.
Classification should be only mentioned, not described in extenso.
Role of CT and DSA should be described in Introduction, since many patients from the cohort had this two types of investigations.
In clinical features potential changes in skin colour should be included.
Case 1: no VM present without Gd enhancement
Case 4- why was sclerotherapy used in suspected high- flow malformsation?
Changes in skin colour are not included in presentations of different VM in Discussion.
Conclusions should be based on study findings- evaluation of the treatment should be performed after each therapy and in a case of failure alternative pathology should be suspected.
Round 2
Reviewer 2 Report
Comments and Suggestions for Authors
Thank you for revision.